# Global Covenant of Mayors, a dataset of GHG emissions for 6,200 cities in Europe and the Southern Mediterranean Countries

Albana Kona[1], Fabio Monforti-Ferrario[1], Paolo Bertoldi[1], Marta Giulia Baldi[1], Georgia Kakoulaki[1], Nadja Vetters[1], Christian Thiel[1], Giulia Melica[1], Eleonora Lo Vullo[1], Alessandra Sgobbi[2], Christofer Ahlgren[2], Brieuc Posnic[3]

[1]European Commission – Joint Research Centre - Ispra
[2]European Commission – Directorate General for Climate Action
[3]European Commission – Directorate General for Energy

*Correspondence to*: Fabio.MONFORTI-FERRARIO@ec.europa.eu

**Abstract**. The Paris Agreement has underlined the role of cities in combating climate change. The Global Covenant of Mayors for Climate and Energy (GCoM) is the largest international initiative dedicated to promote climate action at city level, covering globally over 10,000 cities and almost half the population of the European Union (EU) by end of March 2020. The fifth Intergovernmental Panel on Climate Change (IPCC) report notes that there is a lack of comprehensive, consistent datasets of cities' Greenhouse Gas (GHG) emissions inventories. In order to partly address this gap, we present a harmonised, complete and verified dataset of GHG inventories for 6,200 cities in European and Southern Mediterranean countries, signatories of the GCoM initiative. To complement the reported emission data, a set of ancillary data that have a direct or indirect potential impact on cities' Climate Action plans were collected from other datasets, supporting further research on local Climate Action and monitoring the EU-27 progress on Sustainable Development Goal (SDG) 13 on Climate Action. The dataset is archived and publicly available with the DOI number https://doi.org/10.2905/57A615EB-CFBC-435A-A8C5-553BD40F76C9.

## 1. Background

Cities consume over two-thirds of the world's energy and generate about 70 % of global GHG emissions (IPCC, 2014), while being at the same time particularly vulnerable to the impacts of climate change (Reckien et al., 2018). An increasing number of cities have voluntarily adhered to transnational networks active in Climate Action (Busch et al., 2018; Heidrich et al., 2016; van der Ven et al., 2017). As these networks and initiatives have evolved, cities' ambition and climate targets have increased to match or even go beyond the ambition of countries (Bertoldi et al., 2018d).

However, the scientific community notes the current lack of systemic knowledge of cities' quantified contribution to combating climate change (Acuto et al., 2018; IPCC, 2015). This knowledge gap originates from many issues, including dissimilarities in the methodologies used for developing local emission accountings and reference scenarios and for setting ambition targets, as well as the absence of a global, open and harmonised dataset of cities' emissions inventories (Kona et al., 2018). Only as

recently as in 2019, the first datasets were published in academic literature, aiming to fill regional gaps (Adami et al., 2020; Kilkis, 2019; Palermo et al., 2020b).

The dataset presented in this paper aims to fill these gaps in Europe and Southern Mediterranean countries (Table 1). It consists of a harmonised, comprehensive and verified dataset of GHG emissions based on data produced by 6,200 cities in the EU-27; European Free Trade Association (EFTA) countries and UK; Western Balkans; Eastern Europe and Southern EU neighbourhoods (Figure 1).

The dataset is extremely valuable for communities engaged in climate action, including policy makers at all levels pursuing informed decisions, as demonstrated by the several and diverse studies based on GCoM data that have appeared in literature in latest years. To illustrate, for example (Palermo et al., 2020a) discussed the mitigation policies at the local level, (Pablo-Romero et al., 2018) analysed the so-called Benchmarks of Excellence actions, while (Croci et al., 2016) analysed the major cities present at that time in the database and (Famoso et al., 2015) focused the study on the participation of signatories from Sicily.

Thanks to the improved accessibility of the dataset and the addition of a number of useful ancillary variables, we expect the number of studies to grow in the near future. In order to support further the scientific community, along with the dataset, we provide the method of producing the data, the corresponding metadata, as well as the technical validation performed.

The European Commission launched the Covenant of Mayors (CoM) in 2008 to endorse and support the effort of EU local authorities in mitigating climate change. In 2015, the Covenant expanded to include climate adaptation. In 2011, the initiative was launched in the EU's Eastern and Southern neighbourhoods, and in 2016 the initiative became global, through the launch of the Global Covenant of Mayors for Climate and Energy (GCoM). The initiative registered a very rapid growth from 241 signatories in 2008 in the EU-27 to more than 10,000 signatories covering more than 869 million inhabitants worldwide in March 2020.

The Joint Research Centre (JRC), the science and knowledge service of the European Commission, provides scientific and technical support to GCoM cities in the development and implementation of their Climate Action plans. The scientific support is given through guidance on methodologies for emission accounting and climate adaptation, as well as through the development of urban policy tools for Climate Action (Bertoldi et al., 2018a, 2018b; Kovac et al., 2020; Monforti-Ferrario et al., 2018; Peduzzi et al., 2020). The technical support consists of checking and validating the data reported by cities in the MyCovenant platform (www.covenantofmayors.eu).

The published dataset (Kona et al., 2020) contains verified reported GHG emissions for 6,200 European and South Mediterranean cities for a set of reference years. Given the voluntary nature of the GCoM and the difficulty of local authorities to report using a harmonised framework, a statistical method for checking the reliability, cleaning and validating the reported data was developed and applied. The method allows building a coherent dataset and consists of four steps:

- Reporting principles, data extraction and clustering of signatories into two groups (large /small areas) based on population size and degree of urbanization (threshold 50,000 inhabitants) (see section 2.1);

- Outlier identification and treatment in large urban areas: coherence and completeness checks on the data reported in the platform with the official Sustainable Energy and Climate Action Plan (SECAP) document (see section 2.2);

- Outlier identification and treatment in small and medium towns: statistical method applied for the identification and correction of outliers in small urban areas in small medium towns (see section 2.3);

- Ancillary data: signatories from the EU are matched with their respective administrative units in the EU official statistics for cities. Harmonised statistical information on signatories allows building a referenced structure for collecting, processing, storing, analysing and aggregating data to support the monitoring of the EU progress on the Sustainable Development Goal (SDG) 13 on Climate Action (Eurostat, 2020) (see section 2.4).

The dataset (Table 1 online dataset) thus contains adjusted self-reported data from cities (i.e. GCoM dataset 2019: Emission Inventories) coupled with ancillary data (GCoM dataset 2019: Ancillary data) related to geographic attributes (area, latitude and longitude, local administrative codes, heating degree-days), socio-economic aspects (GDP per capita) and demographic characteristics at city level (degree of urbanisation, population time series). A detailed technical evaluation at the city level was also performed against the independent estimates provided by the Emissions Database for Global Atmospheric Research EDGAR v5.0 (it provides time series of global anthropogenic emissions of greenhouse gases and air pollutants by country on a spatial grid (Crippa et al., 2020).

In compliance with the EU data policy, we are now in a position to share with the community a ten years dataset validated and harmonised with the EU statistical information system of local authorities. The validation process assesses the completeness of the data (i.e. to minimum reporting requirement), the coherence of the data (i.e. data reported in the platform are coherent with the Climate Action plan document) and includes a data cleaning step (i.e. detection of outliers and their treatment). In spite of the overall good quality of the dataset, some limitations and uncertainties remain and are described under the "Limitations and future work" section.

The resulting dataset is of great value and interest and targets the needs expressed clearly by the scientific and academic community and governmental institutions. This is demonstrated through several data release requests that have been received from different groups. These include the IPCC working group on Climate mitigation chapter, the United Nations Framework Convention on Climate Change non-state-actors zone platform for Climate Action (UNFCC-NAZCA), governmental and research institutions of EU-27 Member States interested in the local contribution to the national reduction targets, and other sub national levels interested in understanding the active territorial participation to Climate Action movement.

A distinctive characteristic of the European GCoM initiative is that it includes in its members small towns interested and engaged in Climate Action, often absent from other initiatives. Therefore, this dataset offers cities of all sizes the means to formulate a comparative analysis of the magnitude, efficiency and intensity of energy use and GHG emissions. Users are warned to read in detail the description of the dataset provided in this paper and to be aware of the overall scope of the GCoM initiative in designing their investigations. In particular, they should be aware that the initiative has never meant to be a method

to create exhaustive inventories of all emission sources in the territory or to deal with emissions already included in national-scale control initiatives, such as the EU Emission Trading System (ETS) mechanisms. GHG emissions mainly reported are $CO_2$ emissions, and in rare cases $CH_4$ or $N_2O$. Further to the information presented in this paper, a more comprehensive amount of information and the corresponding guidelines are available through the website.

## 2. Methods

Hereafter we describe the methods used to produce and consolidate the final dataset. Due to local authorities' difficulties in harnessing and reporting data within a harmonised framework, which may differ from the national emission reporting, not all the self-reported data could be considered reliable. Therefore, a method was developed to construct a robust dataset of emission inventories, organised into four steps:

- Step 1: Data reporting principles, extraction and clustering: accounting principles of GHG reporting framework, data extraction and clustering of signatories into two groups (large/small areas) based on degree of urbanization and/or population size (threshold 50,000 inhabitants);

- Step 2: Detection of outliers from large urban areas: Digital curation of data reported in the platform were performed in terms of completeness and coherence with the official Climate Action plan document, the so-called SECAP in large urban areas;

- Step 3: Detection of outliers from small medium towns: statistical method for the identification and detection of outliers in the GHG emission dataset in small medium towns;

- Step 4: Matching emission data with ancillary data: signatories from the EU are matched with their respective local administrative units of the Geographic Information System of the European Commission.

### 2.1 Data principles, extraction and clustering

To streamline measurement and reporting procedures under the GCoM, a Common Reporting Framework (GCoM CRF) was developed during 2018 in consultation with partners and signatories. While the platforms differ in terms of the data collection approach, they are aligned with the GCoM CRF. The dataset provided in the current study is based on the information reported by signatories through MyCovenant platform, one of the officially recognized reporting platforms of the initiative, and the one used by majority of the signatories. Hereafter we report a brief description of the data collected on MyCovenant in alignment with the GCoM CRF.

The reporting framework is built upon the Emission Inventory Guidance, used by the European Covenant of Mayors and the Global Protocol for Community-Scale Greenhouse Gas Emission Inventories (GPC), used by the Compact of Mayors. Both refer to the 2006 Intergovernmental Panel on Climate Change (IPCC) Guidelines for National Greenhouse Gas Inventories. The protocols for accounting the cities' emissions differ mainly in the principles and minimum reporting requirements on sources, the type of gases and the boundary of the inventory to be reported.

The protocol for accounting the emissions is closely aligned with the IPCC 2006 guidelines regarding the source category of the in-boundary emissions (i.e. the administrative boundaries). It includes

"sources" and "activities" rather than the scope framework used in other city protocols (for example the GHG protocol of WRI). Nevertheless, the emission inventory is not meant to be an exhaustive inventory of all emission sources in the territory. It focuses mainly on GHG emissions related to sectors (stationary energy, transport and waste/wastewater) upon which the local authority could intervene through sectoral measures and urban policies. Signatories can report as well GHG emissions from Industrial Processes and Product Use (IPPU) and Agriculture, Forestry and Other Land Use (AFOLU) sectors where these are significant (Table 2).

Moreover, one of the main differences on GHG emission reporting between non-state (e.g. cites) and state actors is the level of flexibility in choosing the inventory year with the most reliable data. The recommended baseline year for reporting is 1990, or the closest subsequent year for which the most comprehensive and reliable data can be provided (for example 2005).

The geographical boundaries of the "local territory" are the administrative boundaries of the entity (municipality, region) governed by the local authority which is a signatory to the GCoM. Regarding the type of gases, GCoM signatories shall report emissions of carbon dioxide ($CO_2$), methane ($CH_4$) and nitrous oxide ($N_2O$) converted into $CO_2$-equivalents ($CO_{2\text{-eq}}$.), according to their global warming potential (Local governments should disclose also which GWP factors they are using). The three main GHG emission categories included in the inventories are:

- Direct emissions due to final energy consumption, excluding those from industrial plants involved in the ETS.

- Indirect emissions related to grid supplied energy (electricity, heat, or cold) consumed in the local territory (Kona et al., 2019).

- Non-energy related direct emissions (such as from waste, wastewater) that occur in the local territory, if the Climate Action plan contains measures to reduce such GHG emissions.

The GHG emissions are automatically derived in the platform as the product of activity data (detailing the energy consumption/waste per carrier/type) and emission factors, as reported by the signatories (Table 2). The emission factors are coefficients, which quantify the emissions per unit of activity, and one out of three approaches can be used:

- IPCC (2006)– emission factors for fuel combustion – default values mostly based on the carbon content of each fuel;

- National or subnational emission factors for fuel combustion when these are different from the IPCC's.

- Life Cycle Assessment (LCA) – emission factors for the overall life cycle of each energy carrier, i.e. including not only the GHG emissions due to fuel combustion but also emissions of the entire energy supply chain – exploitation, transport and processing.

The procedure to verify and improve the coherence of the dataset starts with the extraction of complete emission inventories stored in a relational SQL database. At the closing date of this study, (September 2019) 6,239 Climate Action plans with complete inventories have been submitted by cities in the EU-27, EFTA countries and UK, Western Balkans, Eastern and Southern EU neighbourhoods.

Inventories and other data are self-reported to the online platform and must accurately reflect the content of the official Climate Action plan (called Sustainable Energy and Climate Action Plan (SECAP) document. The SECAP document is a separate file, usually in PDF format and publicly available, that represents the official action plan endorsed and signed by the local council.

The first step to understand the degree to which this is true and the quality of the reported data, yearly GHG emission per capita are plotted for each signatory (Figure 2). The occurrence of outliers (eg. large amounts of per capita values) is a clear indication of errors in the data, therefore not all the data collected in the platform are consistent with the SECAP document. As the calculations of performance indicators for the dataset, such as the mean and standard deviation, can be distorted by a single grossly inaccurate data point, checking and treating outliers is a routine part of data analysis.

Due to the high volume of information, it is not feasible to check individually the consistency of all the data objects with the SECAP document. The collection of the attributes (i.e. the variables: 15 energy carriers and 16 subsectors) describes the data objects (it is also known as record, point, case, sample, entity, or instance), which visually corresponds to the rows in the Excel files.

The original dataset comprises 6,239 signatories with a baseline inventory, out of which 1,845 with an additional monitoring inventory. In each inventory the cities report at maximum data for 15 energy carriers grouped into 16 subsectors, resulting therefore into 1.94 million data objects. The 16 subsectors have been grouped into 6 sectors (i.e. municipal, residential, tertiary, manufacturing and construction industries, transportation and waste sector, see Table 2) and null objects were deleted, leading in total to 61,207 data objects.

We therefore adopted a rule to treat the outliers, based on the benefits expected when scrutinizing the dataset for the overall assessment of the initiative. Data users willing of producing performance indicators on the impact and the contribution of Climate Actions planned and implemented by CoM signatories must benefit of a robust dataset in order to avoid artefacts and unreliable results. In this context, it is evident that, the bigger a city is, the more impact any errors will have on the overall dataset. In order to have an accurate representation, it is then of utmost importance that large cities have highly accurate data. Hence, we decided to adopt a customized method to treat outliers based on the signatories' degree of urbanization and population size (source https://ec.europa.eu/eurostat/web/nuts/local-administrative-units). The 6,239 signatories and their data were clustered into two groups:

- Large urban areas (densely populated area with a population density of at least 1,500 inhabitants per km$^2$ and a minimum population of 50,000): for this group manual curation of imputed errors in inventories was implemented, which significantly increased the performance indicators of the database by increasing their robustness (described in step 2).

- Small towns and rural areas (intermediate and thinly populated areas): for this group an automatic routine to identify and remove the outliers is applied. The rules governing the automatic detection and treatments of the outliers are detailed in Step 3.

## 2.2 Data cleaning – large urban areas

In this section, we describe the steps followed to detect and treat the outliers in inventories from large urban areas (i.e. cities and greater cities, with a population density of at least 1,500 inhabitants per km$^2$ and a minimum population of 50,000) along with correctness and completeness checks in the overall

dataset. The identification and treatment of outliers in this group of cities has been performed qualitatively.

Because of the harmonisation process of GCoM administrative data and local administrative units Eurostat database 2018 (Eurostat, 2018), 430 signatories covering 116.2 million inhabitants, are classified as cities and greater cities. In addition, in the other regions out of the EU-27 (i.e. Eastern Europe; Western Balkans and Southern Mediterranean) where the classification was not available, we adopted as criteria only the population size as the threshold (i.e. a minimum population of 50,000). Hence, within the GCoM 2019 dataset there are 701 baseline inventories presented by large urban areas, covering a total population of 165.26 million inhabitants.

As part of the evaluation process carried out by JRC on individual SECAPs, activity data were compared against the national/ EU averages (available at national/EU statistical systems such as Eurostat, European Environment Agency). In case of reported data that ranged out of one or more units higher than the average of the sectors national average, we double-checked the accuracy of the platform's reported data with the SECAP document. As a result of the digital curation of outliers, identified through the comparison of self – reported data in the MyCovenant platform against the same data declared in the SECAP, twenty inventories (i.e., about 3 %) have been manually corrected. The SECAP document represents the official action plan endorsed and signed by the local council; therefore, we assume as valid the data reported in the SECAP. The errors were often due to the misinterpretation of the unit measure to be reported in the online template (e.g. kWh/year instead of MWh/year, etc.).

At this point in the procedure, with the help of the statistical routine and the digital curation, we have consolidated the dataset related to activity data. The next step consists of comparing the emission factors used in GCoM inventories against the reference values from IPCC AR4 (IPCC, 2007) and the JRC databases (Lo Vullo et al., 2020) and their completeness (i.e. missing data on emissions were derived from reported activity data and vice versa). In case of reported emission factors that ranged out of ± 50 % of the reference value, we corrected them with the corresponding reference value. As a result of this procedure, there were 153 inventories from large urban areas, where 9.7 % (i.e. 526 out of 5,433 objects) of the data objects were corrected.

### 2.3 Data cleaning – small and medium towns

In this section we describe the automatic routine implemented to detect and treat the outliers in inventories from small medium towns (number of inventories = 5,538 covering a total population of 46.78 million inhabitants).

Urban GHG emissions per capita may deviate significantly from national averages, due to the tendency of emissions to concentrate around human activities. Therefore, setting exclusion ranges of outliers in the per capita GHG emissions based on the national averages may lead to the exclusion of a high number of valid emission inventories from the GCoM dataset. To avoid this bias, we apply a statistical method based on intrinsic properties of the distribution of the emissions in the GCoM database. This allows identifying more accurately potentially unreliable emission inventories and the outliers likely to be the results of incorrect data entry.

The procedure starts with dividing the data into two groups based on the normalization process: the activity data in the residential/municipal/institutional/tertiary buildings and transport sector were

normalised with the population size, whereas the activity data in manufacturing and construction industries were normalised with the GDP values. The majority of these industries are already governed by the cap and trade system (ETS), therefore they are not recommended to be reported in the GCoM platform, although exceptions exist. In addition, signatories that report manufacturing emissions are generally large urban areas (80 % of the activity data within this sector is reported by cities and greater cities), which we have been already examined individually to check for outlying data.

The outliers identification method is based on a generalised extreme studentized deviate (ESD) procedure for the detection of abnormal energy consumptions. The ESD is commonly used in literature (Cerquitelli et al., 2019; Gant., 2013; Rosner, 1983; Seem., 2007), because of its excellent performance under a variety of conditions to detect one or more outliers in a dataset that follows an approximately normal distribution. The per capita activity data in the residential/municipal/institutional/tertiary buildings and transport sector follows approximately a normal distribution.

The procedure iteratively identifies the extreme values in the dataset and then selects to remove those observations which are higher than the extreme values with a confidence level of 95 %. A detailed description of the routine is available at Supplementary File 1 and Supplementary File 2.

Applying this approach, 39 inventories were removed from the initial dataset (i.e. from initial 5,538 inventories). These signatories received a further feedback in addition to the routinely checks already performed at the time of data submission  and have been approached to check and correct the data in the online platform. The clean and robust dataset thus contains 5,499 inventories. As a result, the original inventory containing 6,239 entries was reduced to a clean dataset of 6,200 signatories (i.e. 99 % of the original data), referred to hereafter as the "GCoM dataset 2019: Emission Inventories".

To conclude, also a non-parametric statistical procedure, i.e. the Median Absolute Deviation (MAD), has been applied to identify outliers in dataset that do not follow a normal distribution. This method is more robust than the ESD, but less efficient, and its validity increases as data approach a normal distribution. Similar to the ESD, the choice of the critical value is motivated by the reasoning that if observations other than outliers have an approximately normal distribution, it picks up as an outlier any observations more than about three standard deviations from the means. The results of the MAD procedure produce the same outliers as the ESD procedure; therefore, we argue that the assumption on the quasi normal distribution is correct.

The next step consists of verifying the emission factors used in the inventories, against the reference values from IPCC 2006 and the JRC databases (Lo Vullo et al., 2020), and their completeness (i.e. missing data on emission were derived from reported activity data and vice versa). In case of reported emission factors that ranged out of ± 50 % of the reference value, we corrected them with the corresponding reference value. Because of this procedure, there were 3,019 inventories from small towns, where 15% (i.e. 8,008/52,496 records) of the data records were corrected (Table 3)**Error! Not a valid bookmark self-reference.** compares the main descriptive parameters of the two datasets. The main difference can be noted in the skewness parameter. Both frequency distributions have a positive skewness, meaning that the right tail is longer and the mass of the distribution is concentrated on the left of the figure.

### 2.4 Matching emission data with ancillary data

GCoM signatories, when submitting their data to the MyCovenant platform, report the local authority name, the country and their centroids' coordinates. Through these three attributes, we have been able to digitally match the signatories with their corresponding local administrative units in the Geographic Information System of the European Commission (GISCO) (Eurostat, 2018). Harmonised statistical information on signatories allows building a referenced structure for collecting, processing, storing, analyzing and aggregating data.

In this way, we can derive all ancillary data related to institutional, demographic and socio-economic dimensions:

- Institutional dimension: the GCoM signatories are associated with their correspondent NUTS codes; the Local administrative units' codes; their Functional urban area and cities codes; the geographical coordinates; the area and shape files of their local administrative units;

- Demographic dimension: the GCoM signatories are associated with the population data in 2018; the degree of urbanisation;

- Socio-economic and climate dimension: the GCoM signatories are associated with the GDP at NUTS 3 level; and heating degree-days at NUTS 3 level.

The aim of the ancillary data is also to support the monitoring of the SDG 13 on Climate Action in an EU-27 context, which focuses on climate mitigation, climate impacts and on initiatives that provide support to Climate Action, as the GCoM. More broadly, the ancillary data could support further research on investigating drivers of Climate Action at city level and the development of urban policy design. In addition, we extracted the national values of GHG emissions per capita from EDGAR v5.0 for the corresponding GCoM activity sectors (Table 4**Error! Reference source not found.**).

### 3. Data availability

The dataset is archived and publicly available with the DOI number https://doi.org/10.2905/57A615EB-CFBC-435A-A8C5-553BD40F76C9 (Kona et al., 2020).

### 4. Benchmarking

In the case of GCoM, the uncertainty of reported emissions is particularly difficult to estimate since non-formal uncertainty analysis is applied by cities on the activity data and the emission factors. Hence, given this limitation, we argue that the best practical way to assess the uncertainty of reported data is to perform a detailed benchmark of the overall dataset against international emission dataset such as EDGAR v5.0. A similar approach has been applied to validate cities emission data in United States (Nangini et al., 2019).

Although such a procedure does not necessarily implies an absolute validation of our data, could clarify to what extent the dataset is comparable with an internationally reputed source as EDGAR v5.0 is. It is also worth noticing that the two benchmarked datasets are different by principle, as GCoM collects data from local authorities with a supposedly good knowledge of their territory, while the methods used in

EDGAR v5.0 (Crippa et al., 2020) downscale the emissions from a national or subnational scale to finer scales using spatial proxies and present results in gridded maps. EDGAR v5.0 combines several proxies ranging from population density to specific point source location maps for estimating emissions of different economic sectors. Regardless the different approach, the potential use of EDGAR gridded data for the examination of emission in large sample of cities worldwide has been already noticed in literature (Marcotullio et al., 2014).

Using ArcGIS, we overlaid the signatories' urban spatial boundaries onto the EDGAR v5.0 emission grids. We then used the built-in Spatial Zonal Statistics tool to estimate total emissions for each urban area and two source categories: energy in buildings and road transportation. EDGAR v5.0 includes emissions from a variety of sources (Solazzo et al., 2021) at the aggregate level of at least 0.1° spatial resolution (representing about 10 x 10 $km^2$ at the equator). Here we use the EDGAR v5.0 global grids of estimated emissions in metric tons for the year 2005 for the most prevalent GHGs: carbon dioxide excluding short cycle organic carbon (i.e. $CO_2$_excl_short-cycle_org_C). Emissions of $CO_2$_excl_short-cycle_org_C include all fossil $CO_2$ (such as fossil fuel combustion) and exclude all sources and sinks from land-use, land-use change and forestry (LULUCF) (Crippa et al., 2020). Overall, we compared data from 1945 signatories from EU-27 + UK countries with EDGAR v5.0 corresponding data on direct emissions in energy in buildings sector and road transportation.

Annex 1 provides a detailed discussion of the benchmark of the two datasets in both transport and building sectors.

Overall, considering the completely different origin of EDGAR v5.0 and GCoM primary data the agreement has to be considered satisfactory, taking into consideration the well-known difficulties in matching inventories based on top-down or bottom-up approaches and the uncertainties affecting inventories in general (see e.g. (Solazzo et al., 2021) for a deep analysis of EDGAR v5.0 uncertainties).

## 5. Main findings

Local authorities that adhere to transnational networks active on climate action, by making publicly available the plan, without any obligation to do so, render themselves accountable both globally and locally (Gordon, 2016). This paper presents a major attempt to provide the scientific community with a reliable, consistent and complete dataset, derived from the cities' plan submissions. The following provides an overview of the results extracted from the analysis of the dataset in terms of signatories' participation, the submission status of the Climate Action plans, as well as its implementation progress in terms of the emission trend.

Starting with adhesion, Table 3 in the dataset reports the full list of the 8136 signatories and associated ancillary data. The ancillary data comprises institutional (i.e. statistical administrative information), demographic (i.e. population, degree of urbanization) and socio-economic data (i.e. GDP, heating degree-days, national GHG emissions per capita). Harmonised statistical information on signatories (i.e. the ancillary data) allows building a referenced structure for collecting, processing, storing, analysing and aggregating data to support the monitoring of the EU-27 progress on the SDG 13 on Climate Action (Eurostat, 2020).

Regarding action planning, three-quarters of these signatories (i.e. 6200 local authorities) submitted an action plan, comprising a baseline emission inventory and a set of actions to reach their climate mitigation goals.

About the trend, less than one third of these submissions (i.e. 1845 signatories), reported progress on the implementation of the action plan by presenting a second inventory, called monitoring report. Table 2 in the dataset reports the activity data and related emissions mapped in the baseline and monitoring inventories aggregated into Stationary energy, Transport and Waste subsectors.

Hence, the progress made by the signatories in the implementation of their climate action plans is assessed in terms of emissions based on these data. Since the inventories might have different reporting sectors, we analyzed the trend considering only those sectors reported within both inventories, the baseline and the monitoring one. We found that the absolute reductions achieved from baseline inventories to monitoring inventories correspond to 23 %. If cities had progressed linearly towards their target, these signatories would have achieved 17 % of emission reduction by the inventory years, which is lower than the 23 % observed reduction. Consequently, we can assume that monitoring signatories are on track to reach their commitment.

## 6.   Limitations and future work

Despite the data mining and verification process, a few limitations and uncertainties remain in relation to the data quality. To start with, it is important to highlight the fact that the overall quality of the data reported in the platform depends mostly on the city's capacity to gather and report into the harmonised framework of GCoM. The JRC does not correct or adjust the reported data itself, but it is the responsibility of the signatory, on receipt of the feedback analysis from experts, to check and possibly revise its data according to the Climate Action plan. Indeed, we have noted an increasing quality of data reported from cities since 2010, mainly thanks to this feedback-rechecking system. Therefore, the aim of the approach adopted here is not to validate the data as such (they are collected and reported by the signatories), but to guarantee as far as possible the internal consistency and completeness of the data reported in the online platform with the Climate Action plan documents (i.e. the SECAP).

Secondly,  there is a limited knowledge on the methods used by cities in determining the emissions, especially within the transport sector. The aim of the technical validation reported in section 4 is to compare the GCoM dataset against international datasets such as EDGAR, being well aware of the fact that GCoM reports direct observations, whereas EDGAR calculates emissions following a consistent Tier 1 approach based on AD and EF country specific information.

In small-medium urban areas, we assume that local authorities use the territorial approach based on the collected activity data. For these areas, there is a good match with EDGAR v5.0 data, whereas in large urban areas, we note a significant deviation from EDGAR proxies. As already mentioned,  this is probably due, among other factors, to the differences between locally collected data (e.g. on local fleet) and average national information. Moreover, due to the uncertainty on the methodological differences for accounting the emissions, embedded in the nature of the sector, the emissions in this sector can differ widely between cities with similar patterns or sizes.

Regarding waste, the mapping of emissions in this sector has only been added in the last revision of the reporting framework, therefore we expect more data in this sector to become available as cities integrate it in their inventories. For this reason, data from this category were not included in the present study.

Finally, a major source of uncertainty originates from the use of emission factors developed for the national or sometimes even international scale, especially for the electricity and waste sectors. According to our information, signatories often apply the default emission factors provided by JRC and based on very wide scale IPCC Tier 1 approach. Clearly, such a coarse granularity in not always able to catch local peculiarities and features.

On the contrary, deploying city level emission factors for instance for electricity supplied through the grid, taking also into account local renewable energy production, would greatly increase the accuracy of the data.

Future work envisage the possibility of undertaking a comparable analysis also of the data reported by GCoM signatories through the CDP-ICLEI Unified Reporting System, and other recognized/affiliated national and regional reporting platforms with a view to expanding the coverage of a harmonised, complete and verified dataset of GHG inventories at city level.

**Code Availability:** Most data handling in the methods and technical validation was done in MATLAB (available at Supplementary File 1) and Microsoft Excel (Supplementary File 2).

**Acknowledgements:** We are grateful to local authorities who make public their engagement in Climate Action planning, through their participation in the Global Covenant of Mayors initiative. The authors would like to thank European Commission Directorate General for Energy, the CoM Office and JRC colleagues working in the CoM initiative for their support in giving visibility and effectiveness to the effort of cities and local governments in the climate change action. A special acknowledgment to our colleagues Monica Lanzoni for her in-depth review of the statistical analyses, and to Andreas Kontogeorgos for his contribution to English language review.

**Author contributions:** Kona A., Monforti-Ferrario F. and Bertoldi P. designed the research. Kona A. prepared the manuscript. Kona A., Baldi M.G., Kakoulaki G. and Ahlgren C. assembled and prepared the dataset. All the authors revised the methodological steps and the manuscript.

**Competing interests:** The authors declare no competing interests. The views expressed are purely those of the authors and may not in any circumstances be regarded as stating an official position of the European Commission.

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

**Tables**

**Table 1. Datasets and their attributes.**

| GCoM dataset 2019:<br>emission inventories | GCoM dataset 2019:<br>ancillary data |
|---|---|
| - GCoM_ID: identification code of the signatory<br><br>- emission inventory id: identification code of the inventory<br><br>- emission inventory sector: stationary sources (municipal, residential, institutional/tertiary buildings and facilities, as well as manufacturing and construction industries); mobile sources for transportation purposes (i.e. on-road, rail, waterborne navigation and off-road ) and Waste<br><br>- type of emissions: direct emissions from fuel combustion as above described and from waste/wastewater sector; and indirect emissions due to consumption of grid-supplied energy consumption;<br><br>- type of emission inventory: baseline or monitoring inventory<br><br>- inventory year and population in the inventory year<br><br>- activity data and reporting unit: all activity data (i.e. final energy consumption) occurring in stationary sources and mobile sources for transportation purposes within the local authority boundary are reported in the baseline/monitoring inventories.<br><br>- emission factor type: IPPC factor the activity-based approach or Life Cycle Approach<br><br>- GHG emissions and reporting unit: occurring in stationary sources (excluding "energy generation" industries for  and  the ones under the EU- Emission Trading Schemes); occurring in mobile sources for transportation and non-energy related emissions from disposal and treatment of waste and wastewater generated within the city boundary are reported under waste/wastewater sector. | - GCoM_ID: identification code of the signatory<br><br>- signatory name and country code and Covenant Regions<br><br>- year of adhesion and population in the adhesion year and in 2018; signatory adhesion type<br><br>- longitude; latitude; area<br><br>- regional identification code - level 3 (NUTS3): Nomenclature of Territorial Units for Statistics is a geocode standard for referencing the subdivisions of Member States for statistical purposes<br><br>- local Administrative Units identification code: level 1 and  2 (LAU1/ LAU2)<br><br>- Functional Urban Area identification code: (FUA)<br><br>- Cities and greater cities identification code:  CITY ID/GREATER_CITY_ID and the degree of urbanisation;  type of local authority<br><br>- Heating Degree-Days (HDD) and reference year HDD<br><br>- GDP per capita at NUTS3 (average 2010-2018)<br><br>- GHG emissions per capita in GCoM sectors in EDGAR v5.0 and reference years in EDGAR<br><br>- mitigation reduction target 2020 or 2030: reduction target to be achieved through the implementation of the climate action plan by 2020/2030<br><br>- reduction type: absolute or per capita |

The "GCoM dataset 2019: Emission Inventories" are self-reported data in the MyCovenant platform of the GCoM reporting framework. Table 2 of the online dataset reports these emission related data, while Table 1 reports their metadata description. The "GCoM dataset 2019: ancillary data" comprises geographical attributes, socio-economic aspects and demographic characteristics at city level derived from Eurostat (i.e. the statistical office of the European Union) and from the EDGAR v5.0 (i.e. Emissions Database for Global Atmospheric Research). Table 3 of the online dataset reports these ancillary data, while Table 1 reports their metadata description.

**Table 2. Mapping of emission source categories in GCoM reporting framework based on the IPCC 2006 guidance**

| Sectors and subsectors in GCoM reporting framework | | IPCC (ref no.) | Description |
|---|---|---|---|
| Stationary energy | Residential buildings | 1A4b; 1A1 | All activities and related GHG emissions (direct emission from fuel combustion and indirect emission due to consumption of grid-supplied energy) occurring in stationary sources within the local authority boundary are reported. GHG emissions from sources covered by a regional or national Emissions Trading Scheme (ETS), or similar (i.e. industries with thermal energy in input below or equal to 20 MW) when ETS does not exists, are not accounted in the inventory. In addition, "energy generation" industries/facilities are not reported under this sector to avoid double counting with indirect emissions. |
| | Commercial building and facilities | 1A4a; 1A1 | |
| | Institutional buildings and facilities | 1A4a; 1A1 | |
| | Manufacturing, construction industries | 1A1, 1A2; 1A1 | |
| | Agriculture | 1A4c; 1A1 | |
| | Fugitive emissions | 1B1, 1B2 | |
| Transportation | On-road | 1A3b; 1A1 | All activities and related GHG emissions (direct emission from fuel combustion and indirect emission due to consumption of grid-supplied energy) occurring for transportation purposes within the local authority boundary will be reported. NB. the GCoM dataset reported in this dataset does not reflect only the on road fraction of the transport sector, but all the emission in this sector. |
| | Rail | 1A3c; 1A1 | |
| | Waterborne navigation | 1A3d.; 1A1 | |
| | Aviation | 1A3a; | |
| | Off-road | 1A3e; 1A1 | |
| Waste | Solid waste disposal | 4A | Sources related to disposal and treatment of waste and wastewater generating emissions within the city boundary are reported under Waste sector. Where waste/wastewater is used for energy generation, emissions are not reported under this sector to avoid double counting of indirect emission. |
| | Biological treatment | 4B | |
| | Incineration and open burning | 4C | |
| | Wastewater | 4D | |

**Table 3. Statistical parameters of the GHG emissions in the GCoM datasets 2019**

| Parameters | All CoM dataset 2019 | Clean CoM dataset 2019 |
|---|---|---|
| Number of signatories with complete GHG inventories in the Baseline year | 6,239 | 6,200 |
| Total population in the baseline year [Million inhabitants] | 216.61 | 216.25 |
| Mean [tCO$_2$-eq/cap] | 5.18 | 4.69 |
| Median [tCO$_2$-eq/cap] | 4.78 | 4.75 |

| | | |
|---|---|---|
| Standard deviation [tCO$_2$-eq/cap] | 82.35 | 3.68 |
| Skewness [tCO$_2$-eq/cap] | 76.39 | 2.37 |

**Table 4. Mapping of emission source categories with IPCC categories**

| GCoM sector | Sector code | Sector name | Inventory years | Country ISO code |
|---|---|---|---|---|
| Stationary energy/indirect emissions | 1.A.1.a | 1.A.1.a - Public Electricity and Heat Production | 1990-2018 | 52 ISO codes |
| Stationary energy/residential | 1.A.4.b | 1.A.4.b - Residential | 1990-2018 | 52 ISO codes |
| Stationary energy/ Commercial/Institutional | 1.A.4.a | 1.A.4.a - Commercial/Institutional | 1990-2018 | 52 ISO codes |
| Transportation | 1.A.3.b | 1.A.3.b - Road Transportation | 1990-2018 | 52 ISO codes |

**Figures**

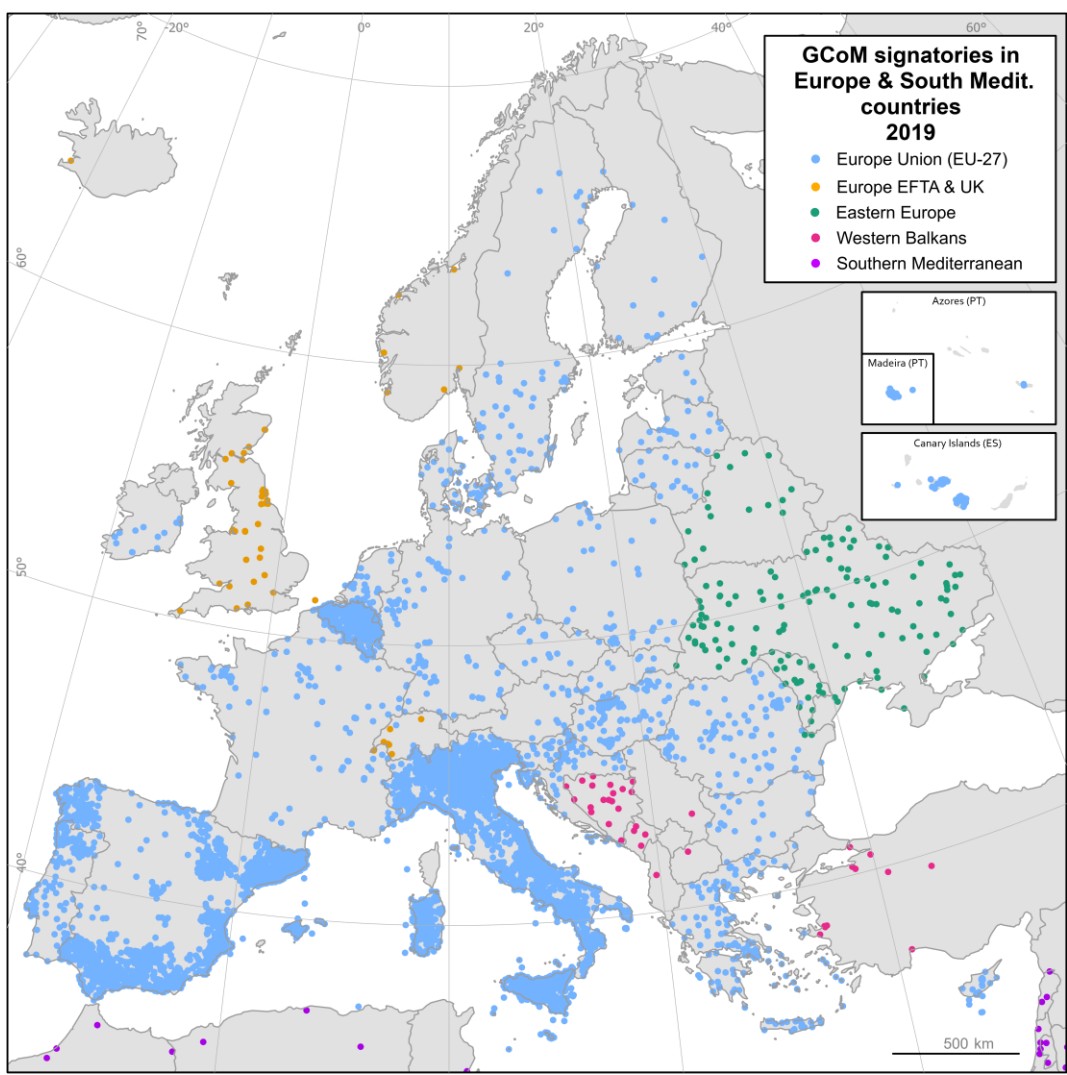

**Figure 1. Global Covenant of Mayors signatories in Europe and South Mediterranean Countries as of 2019.**

The map "GCoM signatories in Europe and South Mediterranean Countries as of 2019" reports the location of the local authorities reporting to MyCovenant platform of the GCoM common reporting framework. Table 3 of the online dataset reports these signatories (8136) and their ancillary data, while Table 1 reports their metadata description. This dataset covers the following countries, grouped into the Covenant region: European Union: EU-27; European Free Trade Association (EFTA) and UK: Iceland, Norway, Switzerland and United Kingdom; Western Balkans: Albania,

Bosnia-Herzegovina, Montenegro, North Macedonia, Serbia and Turkey; Eastern Europe: Armenia, Azerbaijan, Belarus, Georgia, Kazakhstan, Moldova, Tajikistan and Ukraine; Southern Mediterranean countries: Algeria, Lebanon, Jordan, Morocco, State of Palestine, Israel, Tunisia.

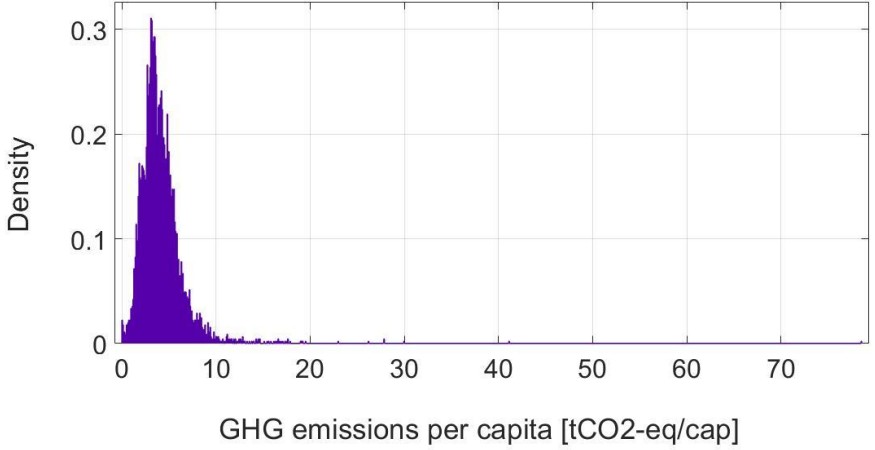

**Figure 2. GHG emissions per capita of signatories in the "GCoM dataset 2019"**

Figure 2 shows the frequency distribution of GHG emissions per capita from emission inventories dataset, with observations that range from 0 to 80 tCO2-eq/cap, with a mean of 5.18 tCO2-eq/cap. These comprehends the emissions from all the GCoM sectors (stationary energy, transportation and waste), excluding manufacturing and construction industries. In the vertical axis the density values are reported, i.e. the share of signatories with the same range of GHG emission per capita, by the width of the range (0.01 in this case).

**Annex 1**

Tables A.1 and A.2 show the values of absolute emissions (A.1) and three selected benchmark indicators – namely correlation, bias and NRMSE- (A.2) for both EDGAR v5.0 and GCoM datasets in road transport and energy in buildings sectors. Values are reported for both the whole set of selected signatories and for three subsets based on population size.

Firstly, it has to be noticed how correlation and NRMSE are consistently better for energy in buildings than road traffic and, for both sectors, tend to improve with increasing the city size, in agreement with the already mentioned fact that the coarse resolution of EDGAR v5.0 limits the description of smaller areas.

Analysing sectors, it is evident as GCoM provides consistently higher values of EDGAR v5.0 for energy in buildings sector, by an average value of 35 % with an overall correlation coefficient of 0.925. This relatively small bias and the good correlation values observed might be attributed to the fact that, in this sector EDGAR v5.0 splits sectorial emissions on the basis of population density, without at the moment considering regional data. Similarly, GCoM signatories collect data from utilities mainly, data that are themselves good proxies of national of energy usage pattern and fuels deployed.

Regarding traffic sector, table A.2 shows also how the bias between EDGAR v5.0 and GCoM emission data for road transport increases in function of the city size, with GCoM transport emissions lower than EDGAR data in smallest cities and overcoming them by a factor of three in largest cities. This is most probably due to the well known fact that largest cities act as traffic attractors: there the number of vehicles is not just proportional to the city population but depends also on the incoming and crossing traffic caused by cities activities, an effect that is captured by GCoM ground based data, but more difficult to be addressed by EDGAR that uses population density as a spatial proxy for emissions allocation. Moreover, the GCoM dataset reported here does not reflect only the on road fraction of the transport sector, but all the emission in this sector, due to the old version of the reporting platform that collected data without distinguishing the modal share. It has also to be considered that GCoM reports real data supplied from the transportation department, which are not necessarily caught in EDGAR: for instance, EDGAR v5.0 uses the average national fleet that could be quite different from a local one and, finally, that the default GCoM emission factors often used by signatories do not apply the very fine categorization of vehicle fleet applied in EDGAR.

On summary, observing the overall values of the benchmark indicators reported in the last line of Table A.2, and considering once again the deep differences in data sources, with GCoM based on local authorities knowledge, while EDGAR v5.0 is based on the use of population density as a main proxy, the behaviour observed does not come as a surprise and confirms the consistency between the two datasets, given the different methodologies applied.

**Table A.1 – Absolute emissions (MtCO₂eq) for road transport and energy in buildings sectors in EDGAR v5.0 and GCoM datasets in selected signatories**

| City Size | Number of signatories | Population in 2005 [million inh.] | Road transport emissions (EDGAR) | Road transport Emissions (GCoM) | Energy in Buildings emissions (EDGAR) | Energy in Buildings emissions (GCoM) |
|---|---|---|---|---|---|---|
| Pop. > 500000 | 8 | 10.3 | 3.366 | 10.691 | 12.598 | 16.022 |
| 50000 < Pop. < 500000 | 96 | 16.3 | 10.909 | 25.097 | 12.525 | 24.566 |
| Pop. < 50000 | 1841 | 9.8 | 19.749 | 12.877 | 12.483 | 13.828 |
| All Sample | 1945 | 36.3 | 34.024 | 48.665 | 37.606 | 54.417 |

660

**Table A.2 – Benchmark indicators for road transport and energy in buildings sectors in EDGAR v5.0 and GCoM datasets in selected signatories**

| City Size | Correlation (Road transport emissions) | Correlation (Energy in Buildings emissions) | BIAS (Road transport emissions) | BIAS (Energy in Buildings emissions) | NRMSE (Road transport emissions) | NRMSE (Energy in Buildings emissions) |
|---|---|---|---|---|---|---|
| Pop. > 500000 | 0.928 | 0.945 | 218% | 32% | 1.481 | 0.449 |
| 50000 < Pop. < 500000 | 0.365 | 0.641 | 130% | 48% | 1.685 | 1.374 |
| Pop. < 50000 | 0.490 | 0.605 | -35% | 10% | 1.536 | 1.411 |
| All Sample | 0.660 | 0.925 | 43% | 35% | 5.168 | 3.336 |