# Peer review of "Global Covenant of Mayors, a dataset of GHG emissions for 6,200 cities in Europe and the Southern Mediterranean Countries"

_Earth System Science Data, 2021_

## Author Comment (AC1)

**Answers to Reviewer #1.**

The authors present and describe a dataset named "Global Covenant of Mayors dataset of GHG emissions for 6,200 cities in Europe and the Southern Mediterranean". The authors do a great job in assembling and maintaining such a huge amount of data and information. If one would only read the abstract, it would perhaps expect a detailed analysis and description of a GHG emission dataset from cities, in the context of the current Paris Agreement and European emission reduction targets, and this is exactly what I was expecting.
However, reading the whole paper, I have the feeling that it misses some substance in the actual meaning and use of this dataset. Even if I understand that this is a dataset description paper and I should not expect much scientific results, somehow I was a bit disappointed not finding the real aim and scope of this study.
My point is, it is an interesting work but I was looking forward to find something about the actual results reported by these 6,200 cities, what is their current state in terms of GHG emissions, and how are they doing in the reporting process and Paris Agreement targets. This is not more than a methodological/abstract statistical paper which, in my personal opinion, would greatly beneficiate of a section summarizing the actual data presented by this dataset. It will give the general audience more insight into what actually this database is about – GHG emissions from cities.
For example one could present 1) a classification of top 10 emitters cities/large areas (in CO2eq or kton gas) and their percentages with respect to the European GHG total emissions highlighting sectors and activities responsible for these emissions (useful for local mitigation strategies); 2) look at emission trends observed between the baseline year (1990) and last available year (2019)? 3) How do local governments make use of this data? Also, authors do not mention other initiatives like e.g. C40 cities.
Overall, I think this paper will be of great interest to all policy/science/industry users and, therefore, I encourage authors to put a little effort and if possible restructure and combine this methodological abstract description with an overview of scientific results.

Response: We would like to thank the reviewer for the relevant comments and good suggestions. We believe the revision performed on the manuscript and the additional section on summarizing the actual data presented by this dataset addresses all the comments and suggestions provided. Detailed response to the comments raised follow.

- We added the following section (in italics) providing an overview of the main results:

*Main Findings*
*Local authorities that adhere to transnational networks active on climate action, by making publicly available the plan, without any obligation to do so, render themselves accountable both globally and locally (Gordon, 2016). This paper presents a major attempt to provide the scientific community with a reliable, consistent and complete dataset, derived from the cities' plan submissions. The following provides an overview of the results extracted from the analysis of the dataset in terms of signatories' participation, the submission status of the Climate Action plans, as well as its implementation progress in terms of the emission trend.*
*Starting with adhesion, Table 3 in the dataset reports the full list of the 8136 signatories and associated ancillary data. The ancillary data comprises institutional (i.e. statistical administrative information), demographic (i.e.*

*population, degree of urbanization) and socio-economic data (i.e. GDP, heating degree-days, national GHG emissions per capita). Harmonised statistical information on signatories (i.e. the ancillary data) allows building a referenced structure for collecting, processing, storing, analysing and aggregating data to support the monitoring of the EU-27 progress on the SDG 13 on Climate Action (Eurostat, 2020).*

*Regarding action planning, three-quarters of these signatories (i.e. 6200 local authorities) submitted an action plan, comprising a baseline emission inventory and a set of actions to reach their climate mitigation goals.*

*About the trend, less than one third of these submissions (i.e. 1845 signatories), reported progress on the implementation of the action plan by presenting a second inventory, called monitoring report. Table 2 in the dataset reports the activity data and related emissions mapped in the baseline and monitoring inventories aggregated into Stationary energy, Transport and Waste subsectors.*

*Hence, the trend on emissions uses these data to assess the progress made by the signatories in the implementation of their climate action plans. Since the inventories have different reporting sectors, we analyzed the trend only in those reporting data within the sector in both inventories, the baseline and the monitoring one. Overall, the trend on emissions from signatories reporting progress shows that the absolute reductions achieved from baseline inventories to monitoring inventories correspond to 23 %. Assuming that cities progress linearly towards their target, these signatories would have achieved 17 % of emission reduction by the inventory years, which is lower than the 23 % reduction. Consequently, we can assume that monitoring signatories are on trach to reach their commitment.*

The manuscript would also beneficiate of a native English speaking review. Therefore, I suggest major revisions and I am really looking forward to an improved version of this manuscript. Below are my comments which hopefully will be appreciated and help in improving the manuscript, followed by a list of line-by-line specific changes:

Response: The manuscript underwent a thorough language check and significant linguistic improvements in the revision phase.

I miss at the end of the title a word, could be cities or countries? I accessed the online database (with more than a click, ESSD data policy states one click to download the data) and I see "countries" after Mediterranean. I would strongly suggest to add it as well to the title.

Response: Thank you for spotting the missing word, we added it in the current version

In general throughout the manuscript, terminology is not always explained. Readers not familiar with this background need to clearly understand the terms and acronyms. Consistency in using acronyms and dataset names is poor. Since the beginning of the paper I stated to get confused by the use of GCoM and CoM, the authors should be consistent and use, in my opinion, only GCoM (online tables as well). I also miss references for many products. Transparency is the key to such data description papers and links to guidelines and regulations need to be provided.

I strongly recommend the introduction of a Figure 1 to define boundaries and study area. Will be helpful to visualize the countries/cities.

*Response: to address it we have converted figure 1 into table 1 with the expanded description of the data field and extensive caption. GCoM and CoM acronyms have been better explained and checked.*

I somehow can't access the Supplementary files by clicking on the link in the manuscript. I could only view them from the preprint ESSDD website.

*Response: the links should work properly now.*

The authors compare GCoM results with EDGAR v5.0 dataset (should be specified everywhere in the manuscript that authors used EDGAR v5.0) They state that "Overall, considering the completely different origin of EDGAR and CoM primary data the agreement has to be considered fully satisfactory".
I personally think that a low uncertainty of two completely different datasets could either imply that both are good or that both are bad. Authors should explain better the methodology used to calculate uncertainties (perhaps add it as an Annex).

*Response: we have rewritten the paragraph clarifying the reason and the methodology used in benchmarking the GCoM dataset with EDGAR v5.0. An Annex has been also added in order to provide numerical details.*

Specific line-by-line comments:

L3: Title: add "countries" after Mediterranean.

*Response: addressed*

L15: Please specify here and throughout the paper what EU are you referring to, I guess it is EU27.

*Response: addressed*

L16: which latest IPCC report? Please add year and reference

*Response: IPCC 2006, addressed*

L17: should be "emission inventory"

*Response: addressed*

L17: I would reformulate: "To partly address this gap, we present a harmonized..."

*Response: addressed*

L18: what do you mean with complete and verified dataset, please explain in a footnote "complete and verified"
As far as I understood, it only contains the three main GHGs (CO2, CH4 and N2O) what makes it complete?. To be mentioned, since the beginning, that you only report these three GHGs.

*Response: addressed as follows:*

*In compliance with the EU data policy, we are now in a position to share with the community a ten years' dataset validated and harmonised with the EU statistical information system on local authorities. The validation process comprises the completeness (i.e. to ensure that minimum reporting requirement are fulfilled), the coherence (i.e. data reported in the platform are coherent with the Climate Action plan document) and the data cleaning (i.e. detection of outliers and their treatment).*

L19: "to complement the reported emission data"

*Response: addressed*

L21: I would call them datasets (and not databases)

Response: addressed

L22: EU (without 's) progress on the Sustainable Development Goal (SDG) 13 on Climate Action (I would add a reference here)

Response: addressed

L22: "The datasets ..." please check the font size and note that this link does not work. Replace it with the one from the Data Availability section or references

Response: addressed

https://doi.org/10.2905/57A615EB-CFBC-435A-A8C5-553BD40F76C9

Actually, the ESSD data policy states that access to data should be done with one click, I already clicked 3 times to download the data.

Response: unfortunately there is no way to avoid the three times of clicking to download the data due to the EC system

L25: I would delete Summary

Response: addressed

L27: Perhaps authors could find a more recent reference?

Response: addressed, the following references have been added

Solazzo, E., Crippa, M., Guizzardi, D., Muntean, M., Choulga, M. and Janssens-Maenhout, G.: Uncertainties in the Emissions Database for Global Atmospheric Research (EDGAR) emission inventory of greenhouse gases, Atmos. Chem. Phys., 21, 5655–5683, doi:10.5194/acp-21-5655-2021, 2021.

L31: dot at the end of the sentence.

Response: addressed

L35: authors state that there is an absence of a global cities datasets, true but other initiatives (e.g. C40 cities) should be mentioned.

Response: addressed.

L39: "gaps for Europe". In the beginning the authors were talking about EU, now is Europe, please use consistent domain.

Response: the use of term Europe is appropriate, as here we are referring not only to the EU-27, but also the eastern Europe, EFTA, UK and Western Balkans.

L40: Mediterranean countries: on L61 is written cities: the authors should be consistent in using the same name of this dataset e.g. countries everywhere.

Response: addressed.

L41: now we have again EU, before was Europe. The Southern EU neighborhoods are the same with the Southern Mediterranean countries? Why not simply EU27 +UK and Western Balkans (I guess non-EU)? What countries are part of the EFTA?

Response: addressed.

As I already mentioned, this paper would really beneficiate of a figure with the study domain, cities/countries.

Response: addressed.

L49: delete space after neighbourhood. It is very abstract, can you please define on the map all these groups? (EFTA, W Balkans, EU27, Eastern and Southern neighborhoods?)

Response: the dataset reports emissions and ancillary data at city level. "Mediterranean countries" term is used as a Covenant region, grouping the cities in that region. Full list of Covenant regions, and the countries pertaining to the regions are explained in table 1, and visible in the cities map (figure 1)

L51: mention that in 2008 we had EU28; please add parties or signatories after 10,000.

Response: addressed

L52: delete as of, in March 2020.

Response: addressed

L55: please add: reporting based on which methodologies? IPCC same as for country reporting?

Response: addressed

L58: The technical report...

Response: addressed

L61: here is cities, on L40 is countries, be consistent. Please state which reference years are you talking about

Response: addressed. The platform gather data with different reporting years. To better clarify, we have added the following.

*Moreover, one of the main differences on GHG emission reporting between non-state (e.g. cites) and state actors is the level of flexibility in choosing the inventory year with the most reliable data. The recommended baseline year for reporting is 1990, or the closest subsequent year for which the most comprehensive and reliable data can be provided (for example 2005).*

L61 and everywhere in the manuscript: CoM and GCom, please use GCoM everywhere as stated by the title. If otherwise, make it clear in the beginning of the manuscript.

Response: addressed

L64: the 4 steps are repeated in the Methods section, I would talk here in more general terms about the dataset and the context and leave the steps to be detailed as done in the methods section

Response: addressed as follows:

- *Reporting principles, data extraction and clustering of signatories into two groups (large /small areas) based on population size and degree of urbanization (threshold 50,000 inhabitants) (see section **Error! Reference source not found.**.1);*
- *Outlier identification and treatment in large urban areas: coherence and completeness checks on the data reported in the platform with the official Sustainable Energy and Climate Action Plan (SECAP) document (see section **Error! Reference source not found.**.2);*
- *Outlier identification and treatment in small and medium towns: statistical method applied for the identification and correction of outliers in small urban areas in small medium towns (see section **Error! Reference source not found.**.3);*
- *Ancillary data: signatories from the EU are matched with their respective administrative units in the EU official statistics for cities. Harmonized statistical information on signatories allows building a referenced structure for collecting, processing, storing, analysing and aggregating data to support the monitoring of the EU progress on the Sustainable Development Goal (SDG) 13 on Climate Action (see section **Error! Reference source not found.**.4).*

L71: to be referenced which method and were is described e.g. see point 2 Methods or Supplement etc.)
Response: addressed

L73: for someone not familiar with this type of terminology, should be explained who are the administrative units in the EU official statistics and link to signatories. Please add footnotes to define each group. Perhaps a list of acronyms would help as well.
Response: addressed, the terminology is fully reported in the table 1, nevertheless they are repeated in the caption

L76: EU progress
Response: addressed

L77: add (SDG), you use it later in the text.
Response: addressed

L78: The datasets (Figure 1): which dataset are we talking about? GCoM? I see now different naming...please explain better the introduction of the "CoM dataset 2019: Emission Inventories" and how it links to the GCoM dataset.
Response: addressed

Figure 1: I would actually have it as Figure 2, as mentioned, Figure 1 should present the domain. Please explain in the figure caption what the acronyms stand for (e.g. NUTS, LAU, GDP, FUA etc.)
Response: addressed

L93: perhaps here you can explain these terms: "complete, cleaned, validated and harmonized"

Response: addressed as follows.

*The validation process comprises the completeness (i.e. to ensure that minimum reporting requirement are fulfilled), the coherence (i.e. data reported in the platform are coherent with the Climate Action plan document) and the data cleaning (i.e. detection of outliers and their treatment).*

L97: Please add example of scientific and academic communities interested in using the dataset. It is important for the local governments and scientist to connect via such resources.

Response: addressed as follows.

*The resulting dataset is of great value and interest and targets the needs expressed clearly by the scientific and academic community and governmental institutions. This is demonstrated through several data release requests that have been received from different groups. These include the IPCC working group on Climate mitigation chapter, the United Nations Framework Convention on Climate Change non-state-actors zone platform for Climate Action (UNFCC-NAZCA), governmental and research institutions of EU-27 Member States interested in the local contribution to the national reduction targets, and other sub national levels interested in understanding the active territorial participation to Climate Action movement.*

L99: Therefore, this datasets offers cities...

Response: addressed

L100: since the beginning you mention GHG emissions: which GHGs?

Response: addressed as follows:

*GHG emissions mainly reported are CO2 emissions, and in rare cases CH4 or N2O*

L107: please include the comments under L64

Response: addressed

L111-L115: Both 2.2 and 2.3 treat the Detection of outliers, 2.2 from large areas and 2.3 from small medium towns, why not call it:

2.2 Detection of outlier from large areas

2.3 Detection of outliers from medium towns

Response: addressed as follows:

*2.2 Data cleaning – large urban areas*

*2.3 Data cleaning – small and medium towns*

L120: Again GCoM appears, please use everywhere CoM or GCoM. Perhaps the authors could present in the introduction some more information on the other platform, CDP-ICLEI Unified Reporting System, why these two platforms? what are the commonalities and differences with My Covenant, why cities should report to one or another?. More insight into methodologies governing these two platforms should be mentioned as well.

Response: We have carefully considered adding further information on the reporting platforms, incl. commonalities and differences, but have concluded that this goes beyond the scope of the paper. The paper only refers to data collected on one of the several platforms that can be used for reporting to the GCoM. The MyCovenant platform currently covers over 90% of GCoM signatories (over 10,000 MyCovenant users vs. below 1,000 GCoM cities reporting to CDP). While they differ in terms of the data collection approach, they are all aligned with the GCoM Common Reporting Framework; hence a description of other differences between the reporting platforms does not contribute to an enhanced understanding of the dataset presented.  To respond to the suggestion, we have made the following amendments and added the following sentences, to reflect on the methodological differences of GHG accounting as practiced by MyCovenant users following the JRC methodology vs. other widely used accounting approaches.

*To streamline measurement and reporting procedures under the GCoM, a Common Reporting Framework (CRF) was developed during 2018 in consultation with partners and signatories. The dataset provided in the current study is based on the information reported by signatories through MyCovenant, one of the officially recognized reporting platforms of the initiative, following the JRC methodology (Bertoldi et al., 2018a). Hereafter we report a brief description of the data collected on MyCovenant in alignment with the CRF. While the platforms differ in terms of the data collection approach, they are aligned with the GCoM Common Reporting Framework.  The dataset provided in the current study is based on the information reported by signatories through the MyCovenant platform. Hereafter we report a brief description of the data collected on MyCovenant in alignment with the GCoM CRF.*
*The reporting framework is built upon the Emission Inventory Guidance, used by the European Covenant of Mayors and the Global Protocol for Community-Scale Greenhouse Gas Emission Inventories (GPC), used by the Compact of Mayors. Both refer to the 2006 Intergovernmental Panel on Climate Change (IPCC) Guidelines for National Greenhouse Gas Inventories.*

L122: The Common Reporting Framework (CRF) is a little bit misleading. CRF is an official  terminology used under the UNFCCC reporting, the tables are named Common Reporting Format, can you please specify in a footnote that CRF is not linked to the UNFCCC process? Readers not familiar with it would think are the same tables.
Response: Thank you for the suggestion. We called it GCoM CRF

L127: Can you give examples of boundaries?
Response: We added territorial

L128: please reference IPCC guidelines, are the 2006 or the update 2019 Refinement or both?
Response: IPCC 2006

L130: give examples please of "other city protocols"
Response: we added the following (for example the GHG protocol of WRI)

L133: replace "they also can report" with "they can report as well.." and who are they? Cities, inventories, protocols?

Response: addressed

Table 1: please replace "with IPCC guidance" with "based on the IPCC 2006 guidelines.
In general, always mention IPCC (2006).
In the column IPCC (ref no.) why 1.A.1 appears in all the rows?
In the Description column, please use capitals for ETS, and delete double spaces. Would be useful to add in the caption which GHGs are we talking about.In general, all captions from this study are very short and not fully explanatory. Waste: to my knowledge, waste sector and activities should be numbered with 5 (5A, 5B etc.).

Response: all comments addressed

L146: The geographical boundaries definition: should be mentioned before, on L127

Response: addressed

L147: "regarding the type of gases" they should be detailed before in abstract and introduction

Response: addressed

L149-150: there are two GWPs largely used depending to which IPCC AR report you refer to, AR4 uses 25 and 298 for CH4 and N2O respectively (and countries report them to UNFCCC CRFs ) and AR5 28/34 and 265/298...which ones are you using??? please define as well GWP in a footnote, you use the GWP 20 or 100 etc.

Response: unfortunately the GWP factor is not reported by the signatories in the reporting platform.

L152: only ETS

Response: addressed

L164: IPCC year and reference

Response: addressed

L169: PostgreSQL database, what is it? And where is described? Please add reference to where can be found (Supplement etc.) or add footnote with explanation

Response: addressed, we modified it as a relational database

L170: Consistency with climate action or Climate Action;

Response: addressed

Is it "had been" the correct tense? these complete inventories were submitted and will never be submitted again? Or could be "have been submitted" because they still exist in present times?

Response: addressed

L173: SECAP appears already at L69, please describe there what it is (add footnote with a link to it if possible)

Response: addressed

L174: add comma after available

Response: addressed

L175-178: these two sentences are repeating, can you please merge and make one sentence? Correct with "reported data, GHG emissions"

Response: addressed

L178: replace with: Figure 2 shows the frequency distribution of 2019 GHG emissions per capita from emission inventory datasets, .."

Response: addressed

L180-183: Could authors please rephrase the explanation for Figure 2? I don't really understand the density here and what exactly is intended with this figure? What is the width of the class, where was defined? Are too many i.e. which interrupts the readability of the sentence.

Response: addressed. We added the following caption to figure 2

*Figure 2 shows the frequency distribution of GHG emissions per capita from emission inventories dataset, with observations that range from 0 to 80 tCO2-eq/cap, with a mean of 5.18 tCO2-eq/cap. These comprehends the emissions from all the GCoM sectors (stationary energy, transportation and waste), excluding manufacturing and construction industries. In the vertical axis the density values are reported, i.e. the share of signatories with the same range of GHG emission per capita, by the width of the range (0.01 in this case).*

L200: please add after: waste sector, see Table 1)

Response: addressed

L218: 2.2 as mentioned before, this section is also about detection of outliers from large urban areas, for a better flow of information I would: - rename the section as "detection of outliers from large urban areas" - add two subtitles Activity data and Emission factors.

Response: addressed, we renamed the sections without a further division.

L225: Eurostat database, please add version, year, reference.

Response: addressed, we added the refence

L225: I think is "are classified as"

Response: addressed

L232: please name some EU statistical systems

Response: addressed.

*(such as Eurostat, European Environment Agency).*

L233: "In case of reported data that ranges out of"

Response: addressed

L234: "accuracy of the platform's reported data"

Response: addressed

L237: "have been manually corrected"

Response: addressed

L238: "therefore, we assume as valid the data reported in the SECAP document."

Response: addressed

L243: IPCC AR4 (and reference) perhaps add a link to a chapter annex etc. So, this means that you are using AR4 GWP values as well? see my notes on line 149, define it there please.

Response: addressed, as mentioned before signatories do not report the GWP value used.

L249: rename with: detection of outliers from small medium towns (see also comments on lines 108-115) I would add here as well Activity data and Emission factors sub-sections to separate better the discussion.

Response: we renamed the sections without a further subdivision.

L264: I think CAP is with capitals and please define it. Again consistency needed throughout the entire manuscript with EU-ETS, or ETS only or acronym versus EU-no acronym etc.

Response: addressed

L265: could you please name these few exceptions in (  )?

Response: for example, out of the EU-27, the other countries mentioned do not have emission trading scheme, therefore in some cases they do include these emissions.

L269-274: Does this method is mentioned before on L202 as the rule to treat the outliers? Of yes, please explain it there.

Response: addressed

L277 and everywhere else: should Supplementary files 1 and 2 be accessible with one click from the text?

Response: Thank you for the suggestion, but unfortunately we cannot modify the system of uploading data with the EC data portal

L282: Now we have a new name of the dataset: "CoM dataset 2019: Emission Inventories" Is this the same with GCoM datasets of 6200 cities from the title? Is this study about this dataset? Is yes, please clarify this in the beginning, as it creates really creates confusion. We have now CoM and GCoM and "CoM dataset 2019" only and "CoM dataset 2019-Emission Inventory" etc. It actually appears on L78 but in this paragraph you state that hereafter you will name it as such.

Response: addressed, they are all rename GCoM now

L285: approaches, Similar to

Response: addressed

L286: .."is motivated by the fact that observations..."

Response: addressed

L292: IPCC 2006 and reference

Response: addressed

L294: be consistent in the paper with space between the value and %, ±50 %

Response: addressed

L295: "the reference value should be (IPCC 2006/JRC)"

Response: addressed

Table 2: Please add a more explanatory caption. The baseline year refers to 1990? should be mentioned.

Response: the baseline years are not fixed at 1990. We have added the following to clarify.

*One of the main differences between non state actors (eg. cities) and the state ones is related to the flexibility of choosing the inventory year with the most most reliable.*

L305: submitting their data to the platform

Response: addressed

L320: Please introduce SDG acronym before (line 22) and be consistent again with the Climate Action in capitals or not, also L322

Response: addressed

L322: now is Global Covenant of Mayors, why not as before, GCoM?

Response: addressed

L324: "we extracted the national values of GHG emissions per capita from EDGAR for the corresponding CoM activity sectors (Table 3)." Please add v5.0 to EDGAR. When you say you extracted the national values of GHG emissions per capita I would expect to see emission values reported in the table, you only present source categories which are not clearly defined in the columns, not knowing which ones belong to which dataset.

Response: addressed

Table 3: this table needs a better caption and explanation as one would not understand where is EDGAR v5.0? and what is the aim of this comparison? This mapping of emission source categories belong to which dataset?

Response: addressed

L335: I would write: 4. "Comparison with ancillary emission inventories"

Response: addressed

L337: This paragraph introduces already the uncertainty subject before the comparison is actually discussed, I would first present the comparison and after discuss uncertainty.

Response: Thank you for the suggestion, addressed

Everywhere: add EDGAR v5.0 please and references.

Response: addressed

L345: when comparing GCoM to EDGAR, how about proxies used by GCoM compared to those in EDGAR? Could you summarize them in a table? This will make a clear discussion on why these datasets are completely different.

Response: Thank you for the suggestion, we believe that we have already provided enough description of

GCoM and their correspondence to the EDGAR sectors.

L349: explain LAU please; delete "s" from emissions (EDGAR emission grids)

Response: addressed

L350: "and two source categories" delete for

Response: addressed

L351: "energy in buildings (RCO)". Where was this defined? I do not find this sector in Table 3, for comparison purposes. Please explain where RCO is coming from, it is an EDGAR code? Why do you use code only for RCO and not for road transportation? I would delete RCO

Response: addressed

L351: This general sentence: "EDGAR includes emissions from a variety of sources" it really needs a reference, examples etc.

Response: addressed.

L352: superscript for km2

Response: addressed

In this paragraph also subscript for CO2

Response: addressed

L358: before on line 352 you used RCO, please be consistent, also with capitals (energy vs Energy)

Response: addressed

L367: here you use " " for the Energy in building(s) sector, and singular for building

Response: addressed

L367: is there a methodology for uncertainty calculation and what exactly this uncertainty represents? Very low values given that datasets are so different

Response: this paragraph was rewritten, please see the new version as well as Annex 1

L368: whereas without capital W

Response: addressed

L372: "emissions in the Covenant framework cover"

Response: addressed

L375: "emissions"

Response: addressed

L376: Further limitations are discussed in the next section

Response: addressed

L377 paragraph: I would still like to know more about the methodology used to calculate uncertainties

Response: Annex 1 has been added

L390: I would use Secondly, You state that "there is a limited knowledge on the methods by cities in determining the emissions..." this is exactly why methodology should become mandatory to GCoM submission system, parties must offer transparency when calculating their emission stating the guidelines, methodology and protocols they use.

Response: addressed

L393: Can you please explain here the word "modeled" ? I am a modeler and I am familiar with the EDGAR database since a long time, I would not name it modeled, It calculates emissions as EM = AD x EF a simple global Tier 1 consistent approach for all countries, but not modeled.
"whereas EDGAR calculates emissions following a consistent Tier 1 approach based on AD and EF country specific information"

Response: addressed

L393: replace suppose with assume

Response: addressed

L393: based on collected activity data

Response: addressed

L394: Replace Indeed with: "For these areas, there is a good match ..."

Response: addressed

L397: Please replace modeling exercise with "emission calculation methodology"

Response: addressed

L408: delete "also"

Response: addressed

L408-410: Will be of great interest to find out more about CDP-ICLEI platform and I am still interested to know why are two systems and how do they differ.

Response: The paper only refers to data collected on one of the several platforms that can be used for reporting to the GCoM. While they differ in terms of the data collection approach, they are all aligned with the GCoM

Common Reporting Framework; hence a description of other differences between the reporting platforms does not contribute to an enhanced understanding of the dataset presented.

L421: our colleague; replace deep reviewing with "in-depth review"

Response: addressed

**Answers to Reviewer #2**

Overall, this is a useful dataset and the work deserves a place in ESSD. I have a number of comments.

Firstly, there are numerous problems with language, which I'm sure ESSD's copy-editor will correct. However, passing the manuscript by someone with good English before submission would make the reviewers' job much easier. There are many situations where the language leaves the meaning unclear.

While it is several times mentioned in the introduction that this dataset is useful, and that the lack of such data has previously been noted (not "denoted"), the introduction does not say how these data would be useful. The fact that data do not exist is insufficient argument for their utility, and it is insufficient to say the data are "extremely useful" or "of great value and interest" without saying how and why. It seems that I wait until section 2.4 on page 11 to find what some potential uses would be, namely "support further research on investigating drivers of climate action at city level and the development of urban policy design." But that only appears to apply to the 'ancilliary data', leaving this reviewer with the belief that the authors don't actually see a use for the main data presented in the article. This cannot be the case. Please make the case for the utility of this dataset much clearer in the introduction of the article.

Response: we would like to thank you for all the comments and suggestions you made in reviewing this paper. Please find in the following the list of topic that underwent a major revision as a follow up to the comments made by you.

We believe that the dataset that we present here coupled with the ancillary data is of great value and interest and responds to the clear needs expressed by the scientific and academic community and governmental institutions. Furthermore, this is the first time that this data set has been used in official statistics thanks to the harmonization process with the statistical system of local authorities in the EU. The dataset used in official statistics is one of the indicators in the Sustainable development Goal (SDG) 13 on climate action at EU level

We added the sections (in italics) to address the comments.

*The CoM dataset has been used for several analyses in the past and we expect the improved dataset accessibility discussed in this paper will allow further use of the data. We have clarified the point early in the paper (Section 1) and added the following references, providing examples of analyses based on CoM data:*
*Peduzzi et al. (2020) Impacts of a climate change initiative on air pollutant emissions: Insights from the Covenant of Mayors, Environment International, Volume 145, December 2020, 106029*

*Palermo et al, (2020) Data on mitigation policies at local level within the Covenant of Mayors' monitoring emission inventories, Data in Brief, Volume 32, October 2020, 106217*
*Pablo-Romero et al., (2017) Analyzing the effects of the benchmark local initiatives of Covenant of Mayors signatories, Journal of Cleaner Production, Volume 176, 1 March 2018, Pages 159-174*
*Famoso et al. (2015) Analysis of the Covenant of Mayors Initiative in Sicily, Energy Procedia, Volume 81, December 2015, Pages 482-492*
*Croci et al. (2017) Urban CO2 mitigation strategies under the Covenant of Mayors: An assessment of 124 European cities, Journal of Cleaner Production, Volume 169, 15 December 2017, Pages 161-177*

The dataset presented excludes some important components. This is stated in the article, but this must be made much more salient, since many readers do not read every word of the data description article. The first time this is mentioned appears to be on page 4, line 131: "the emission inventory is not meant to be an exhaustive inventory of all emission sources in the territory." Then later, at lines 152-3 the reader learns that emissions from "industrial plants involved in Emissions Trading Systems" are excluded. This in fact substantially reduces the utility of the dataset, because it must be combined with other datasets before it can present a comprehensive picture of emissions within a municipality. If these datasets are not going to be combined by the authors, then some clues on how to do this would be of significant use to the potential users of this dataset.

Response: The reader is now warned on the database limitations very early and more emphasis is added to the information. Users are also invited to consult the overall documentation of the CoM initative (e.g., guidelines) available on the website.

When presenting the histogram of per capita emissions, the text implies that there is at least one municipality with zero emissions (line 170 "observations that range from 0..."). Some comment on how it is possible that a municipality can have zero emissions is warranted. Or if this is simply rounded down from some non-zero number, then please provide more precision to the number. It seems highly unlikely that a municipality could have zero emissions.

Response: thank you for the suggestion, indeed it is rounded from some non-zero, and this is due to the fact that some municipalities have reported only data on "institutional buildings", and because the normalization is based on population, the values therefore are low.

Lines 204-5: This is unclear. The statement is that the authors are making performance indicators on the impact and the contribution of climate actions planned and implemented by CoM signatories." But in what follows I cannot see such performance indicators. It seems rather than the assessments being made are informed by the potential uses of the dataset in assessing policy impact. Please clarify.

Response: addressed, we have added the following section summarising the main findings

5.      *Main findings*

*Local authorities that adhere to transnational networks active on climate action, by making publicly available the plan, without any obligation to do so, render themselves accountable both globally and locally (Gordon, 2016). This paper presents a major attempt to provide the scientific community with a reliable, consistent and complete dataset, derived from the cities' plan submissions. The following provides an overview of the results extracted from the analysis of the dataset in terms of signatories' participation, the submission status of the Climate Action plans, as well as its implementation progress in terms of the emission trend.*

*Starting with adhesion, Table 3 in the dataset reports the full list of the 8136 signatories and associated ancillary data. The ancillary data comprises institutional (i.e. statistical administrative information), demographic (i.e. population, degree of urbanization) and socio-economic data (i.e. GDP, heating degree-days, national GHG emissions per capita). Harmonised statistical information on signatories (i.e. the ancillary data) allows building a referenced structure for collecting, processing, storing, analysing and aggregating data to support the monitoring of the EU-27 progress on the SDG 13 on Climate Action (Eurostat, 2020).*

*Regarding action planning, three-quarters of these signatories (i.e. 6200 local authorities) submitted an action plan, comprising a baseline emission inventory and a set of actions to reach their climate mitigation goals.*

*About the trend, less than one third of these submissions (i.e. 1845 signatories), reported progress on the implementation of the action plan by presenting a second inventory, called monitoring report. Table 2 in the dataset reports the activity data and related emissions mapped in the baseline and monitoring inventories aggregated into Stationary energy, Transport and Waste subsectors.*

*Hence, the trend on emissions uses these data to assess the progress made by the signatories in the implementation of their climate action plans. Since the inventories have different reporting sectors, we analyzed the trend only in those reporting data within the sector in both inventories, the baseline and the monitoring one. Overall, the trend on emissions from signatories reporting progress shows that the absolute reductions achieved from baseline inventories to monitoring inventories correspond to 23 %. Assuming that cities progress linearly towards their target, these signatories would have achieved 17 % of emission reduction by the inventory years, which is lower than the 23 % reduction. Consequently, we can assume that monitoring signatories are on trach to reach their commitment.*

The headings of sections 2.2 and 2.3 do not seem appropriate to their contents.

Response: addressed as follows:
*2.2 Data cleaning – large urban areas*
*2.3 Data cleaning – small and medium towns*
Various statements are made about corrections to the data, but only in specific case (lines 279-80) is it made clear that such corrections are fed back to the reporting parties. One is left with the impression that all other 'errors' discovered are not reported back. Please clarify.

Response: Material errors in the submission of data are checked and fed back to reporting parties in the process of data submissions. Nevertheless, after applying the procedure described in the paper, 39 "outliers" inventories

were identified and the related signatories were contacted for further clarifications in order to catch possible errors or inconsistencies not detected by the first screening. The paper was changed to clarify this point.

Line 306: A centroid is a weighted centre based on all points within an area. It seems very unlikely that each of the 6000 municipalities would report a geographic centroid, but rather the coordinates of their town square or post office or similar. Many will not know what a centroid is. Were parties really required to report their 'centroid' and do the authors really believe that centroids were reported?

Response: the centroid data is derived from the Eurostat geographical information. Cities also report their coordinates, but in the dataset we have reported the Eurostat data.

Line 362: While the R-squared is unitless, the RMSE is not. What is the unit of the RMSE here?

Response: this paragraph was rewritten, please see the new version as well as Annex 1

Lines 367-70: I cannot make sense of these sentences. You say uncertainty is low, and then say this could be because EDGAR does it one way while CoM does it a different way, which could only be a supporting argument for uncertainty being high.

Response: this paragraph was rewritten, please see the new version as well as Annex 1

Line 371: Why have the R-squared values been ignored in this analysis, and only RMSEs used? The R-squared for transport is quite poor at 0.66, while the RMSEs (with unknown units) seem both to be relatively good. While 11% sounds low, 0.66 does not sound like low uncertainty, nor does 0.66 seem only "slightly" lower than 0.92.

Response: this paragraph was rewritten, please see the new version as well as Annex 1

Line 372: "the Covenant framework covers only the urban fraction", but given that you've used GIS to isolate EDGAR emissions in the urban areas, so does the EDGAR emission you're comparing with cover only the urban fraction. Please clarify.

Response: Although GIS have helped id us in selecting urban areas, usually GCoM signatories do not include in their inventories emissions arising from roads of national interest, such highways and large roads, even when present in their territory.

Lines 377-8: The authors conclude from this analysis that the agreement between the CoM dataset and EDGAR is "fully satisfactory", which is for me very difficult to agree with. The poor reported correlation for transport emissions actually suggests that one potential use of this dataset might be to indicate that EDGAR's simplistic

methods for spatializing transport emissions need to be improved. A 0.66 correlation does not seem 'fully satifactory'.

Response: The agreement between the GCoM dataset and EDGAR has been put in context and the analysis deeply revised.

Line 396: Additional detail is provided here that should have been presented in the analysis in the previous section, not introduced in the conclusions.

Response: Part of the material has been moved in the previous section, as suggested.

---

## Author Response (AR2)

**Answer to question.**

Reviewer#2

I thank the authors for the significant improvements made to the MS.
One last question: Why is Libya included in the list of countries, but not shown on the map in Figure 1? If this is relatively easy to fix, I'm sure the Libyan participants would be pleased with a revised version

We have checked there are not at the moment signatories from Lybia. We then removed Lybia from the list of countries. The list after Figure 1 has been also rechecked and it is now fully consistent with the dataset.